# Subclinical Myocardial Dysfunction in Type 2 Diabetes Mellitus: Insights from Left Ventricular Diastolic Function and Global Longitudinal Strain Assessment

**DOI:** 10.3390/medsci13040237

**Published:** 2025-10-21

**Authors:** Thao Phuong Nghiem, Hoang Minh Tran, Dung Ngoc Quynh Nguyen, Liem Thanh Dao, Cuong Cao Tran, Tuan Minh Vo

**Affiliations:** 1Faculty of Medicine, Pham Ngoc Thach University of Medicine, Ho Chi Minh City 7000, Vietnam; thaonp@pnt.edu.vn; 2School of Medicine, University of Medicine and Pharmacy at Ho Chi Minh City, Ho Chi Minh City 7000, Vietnam; dr.trancaocuong@ump.edu.vn (C.C.T.); vominhtuan@ump.edu.vn (T.M.V.); 3Faculty of Medicine, University of Health Sciences, Vietnam National University, Ho Chi Minh City 7000, Vietnam; nnqdung@medvnu.edu.vn (D.N.Q.N.); dtliem@medvnu.edu.vn (L.T.D.)

**Keywords:** type 2 diabetes mellitus (T2DM), left ventricular diastolic dysfunction, global longitudinal strain (GLS), subclinical myocardial dysfunction, speckle-tracking echocardiography

## Abstract

Background/Objectives: Diabetic cardiomyopathy in type 2 diabetes mellitus (T2DM) often progresses silently, manifesting as diastolic dysfunction or subtle systolic impairment despite preserved ejection fraction (EF). Detecting these changes early is critical to prevent symptomatic heart failure. This study assessed the prevalence of left ventricular (LV) diastolic dysfunction and impaired global longitudinal strain (GLS) in T2DM patients with preserved EF and identified related risk factors. Methods: We performed a cross-sectional study of 232 adults with T2DM and EF > 50% at a tertiary hospital. Standard transthoracic and speckle-tracking echocardiography were used to evaluate LV diastolic function and GLS. Logistic regression identified predictors of myocardial dysfunction. Results: LV diastolic dysfunction was found in 53.9% of patients, while 13.4% showed impaired GLS (>–17.9%). Independent predictors of diastolic dysfunction were age ≥ 60 years (OR = 2.51, 95% CI: 1.25–5.07, *p* = 0.010) and diabetes duration of 5–10 years (OR = 3.06, 95% CI: 1.46–6.40, *p* = 0.003). Reduced GLS was independently associated with male sex (OR = 2.45, *p* = 0.040) and the presence of diastolic dysfunction (OR = 3.14, *p* = 0.010). Conclusions: Subclinical myocardial dysfunction is common in Vietnamese T2DM patients with preserved EF. Both diastolic dysfunction and reduced GLS may occur independently or together, influenced by age, sex, and diabetes duration. Incorporating GLS into echocardiographic evaluation may enhance early detection and support tailored cardiovascular risk management in this high-risk group.

## 1. Introduction

Type 2 diabetes mellitus (T2DM) has emerged as a major global health burden, contributing significantly to rising rates of cardiovascular disease and premature mortality [1]. Among the cardiac complications of T2DM, diabetic cardiomyopathy (DCM) is increasingly recognized as a distinct clinical entity, characterized by myocardial structural and functional alterations that develop in the absence of coronary artery disease, hypertension, or valvular pathology. DCM often progresses silently, with patients remaining asymptomatic until the disease reaches an advanced stage marked by overt heart failure [2,3].

One of the earliest cardiac abnormalities observed in individuals with DCM is diastolic dysfunction, which typically occurs despite a preserved left ventricular ejection fraction (LVEF) [4]. This condition arises from impaired ventricular relaxation and increased chamber stiffness, eventually leading to elevated filling pressures and heart failure with preserved ejection fraction (HFpEF) [5]. Notably, epidemiologic studies have consistently demonstrated that T2DM substantially increases the risk of HFpEF compared to the non-diabetic population [6]. Importantly, in diabetic patients, HFpEF frequently coexists with subtle systolic dysfunction that may not be captured by standard measures such as LVEF [2,7].

The pathophysiological mechanisms contributing to myocardial dysfunction in T2DM are multifactorial. Chronic hyperglycemia promotes the formation of advanced glycation end-products (AGEs), oxidative stress, mitochondrial dysfunction, and systemic inflammation [8,9]. These processes particularly affect the subendocardial myocardial fibers, which are responsible for longitudinal contraction. As a result, early myocardial changes may remain undetectable with conventional echocardiographic techniques, necessitating more sensitive modalities.

Speckle-tracking echocardiography has emerged as a valuable tool for the early detection of subclinical myocardial impairment in T2DM. By quantifying global longitudinal strain (GLS), this technique assesses myocardial deformation, offering a more sensitive marker of early systolic dysfunction than traditional LVEF [10]. Several studies have shown that reduced GLS is associated with increased risk of adverse cardiovascular outcomes in patients with diabetes, even in the absence of overt clinical symptoms [11,12].

Despite extensive international research, data on subclinical myocardial dysfunction in Vietnamese patients with T2DM remain limited. Existing studies in Vietnam have predominantly focused on symptomatic patients or those with reduced LVEF, leaving a knowledge gap regarding earlier stages of cardiac involvement. Furthermore, the influence of demographic and clinical variables—including age, sex, duration of diabetes, blood pressure, metabolic parameters, and biomarkers such as HbA1c—on myocardial function in this population is not well defined.

Therefore, this study aimed (1) to determine the prevalence of LV diastolic dysfunction and abnormal GLS in T2DM patients with preserved LVEF (≥50%), and (2) to identify the clinical and paraclinical factors associated with these forms of myocardial dysfunction. Elucidating these associations may enhance the early detection of latent myocardial impairment and inform preventive strategies to mitigate progression to overt heart failure.

This study adds to the existing body of knowledge by providing one of the first comprehensive assessments of both diastolic function and GLS in Vietnamese patients with T2DM and preserved EF. By identifying demographic and clinical correlates of early myocardial dysfunction, this work contributes region-specific data that may inform tailored prevention and management strategies in diabetic populations.

## 2. Materials and Methods

### 2.1. Study Population

This is a cross-sectional study aimed at determining the prevalence and associated factors of LV diastolic dysfunction and abnormal GLS in patients with T2DM who have preserved LVEF.

Adults aged ≥18 years with a confirmed diagnosis of T2DM (based on ADA 2024 criteria) [13], in sinus rhythm, and with preserved LVEF (≥50%) assessed by 2D echocardiography using the biplane Simpson’s method.

Exclusion criteria included: patients with type 1 diabetes mellitus; moderate to severe valvular heart disease; diagnosed or suspected cardiomyopathy; active coronary artery disease; arrhythmias other than sinus rhythm (including atrial fibrillation and pacemaker implantation); and clinical, paraclinical, or echocardiographic evidence suggestive of congestive heart failure.

The sample size was estimated based on an expected prevalence of LV diastolic dysfunction of approximately 50% in T2DM patients with preserved EF, as reported in previous studies. Using a precision level of 6.5%, 95% confidence interval, and accounting for a 10% non-response rate, the minimum required sample size was 220 patients. We recruited 232 participants to ensure adequate statistical power for multivariate analysis.

This study was reviewed and approved by the Ethics Committee of City International Hospital, Ho Chi Minh City, Vietnam (Approval No. 56/2024/CIH-HDDD; approved on 20 February 2024). Written informed consent was obtained from all participants prior to their enrollment in the study, in accordance with the Declaration of Helsinki.

### 2.2. Echocardiographic Assessment

Conventional and 2D speckle tracking echocardiography was performed using GE Vivid ^TM^ T8 machines (GE Healthcare Technologies, Inc., Chicago, IL, USA), with images taken during end-expiration in the left lateral decubitus position. Echo images were obtained at over 55 frames per second, and echocardiography was conducted by two physicians with over 5 years of experience in echocardiographic studies. To evaluate intra- and inter-observer variability, a random subset of 20 patients was re-analyzed independently by the two physicians, blinded to clinical data and to each other’s results.Diastolic function assessment (ASE/EACVI 2016):

LV diastolic function was evaluated using the recommendations of the 2016 ASE/EACVI guidelines. Initial classification was based on the mitral inflow E/A ratio and early filling velocity. For patients with intermediate values, three additional parameters were considered: the average E/e′ ratio, tricuspid regurgitation velocity, and left atrial volume index. The integration of these indices allowed grading into normal relaxation, impaired relaxation with normal filling pressures, or elevated filling pressures (grade II–III diastolic dysfunction) [14].
LV geometry:

Left ventricular geometry was categorized according to 2016 ASE/EACVI criteria using the LV mass index (sex-specific cutoffs) and relative wall thickness. Patients were classified as having normal geometry, concentric remodeling, concentric hypertrophy, or eccentric hypertrophy [15].
GLS Measurement:

The LV GLS value is obtained by averaging regional myocardial strain across three views: the apical 4-chamber, 3-chamber, and 2-chamber views. Image selection and endocardial border tracing techniques were performed according to ASE/EACVI 2015 guidelines [16]. Speckle-tracking echocardiography images were processed using AFI 3.0 software provided by GE. An abnormal GLS is defined as a value greater than −17.9% [17].

### 2.3. Statistical Analysis

Stata statistics version 19 was used for statistical analysis. Continuous variables were tested for normality using the Kolmogorov–Smirnov test, presented as mean ± SD, or median, and qualitative variables as frequency (%). To compare groups with normally distributed continuous variables, we used *t*-tests and ANOVA. Non-normally distributed continuous variables were analyzed using Mann–Whitney U and Kruskal–Wallis tests, and the Chi-squared test was used for qualitative variables. To identify predictors for the GLS decline and LV diastolic dysfunction, we conducted the lasso technique in parallel with assessments by AIC (Akaike Information Criterion), BIC (Bayesian Information Criterion), and multicollinearity (VIF) to select the multivariable logistic model.

## 3. Results

### 3.1. Baseline Characteristics

Of the 232 patients who met the inclusion criteria, 46.1% were male. The mean age of the study population was 65.5 years, with a median duration of type 2 diabetes mellitus of 7 years. Comorbid conditions were highly prevalent, most notably hypertension (75.9%) and obesity (39.2%). Dyslipidemia and smoking, both established cardiovascular risk factors, were relatively well controlled, with respective prevalence rates of 16.8% and 9.5% (Table 1).

Echocardiographic characteristics of the study population are presented in Table 2. Left ventricular diastolic dysfunction was observed in a substantial proportion of patients, with a prevalence of 53.9%, while left ventricular hypertrophy was present in 26.7% of cases. The mean global longitudinal strain (GLS) was −19.5 ± 2.5, and impaired GLS (defined as GLS > −17.9%) was identified in 13.4% of the population.

### 3.2. Risk Factors Associated with LV Dysfunction in T2DM Patients with Preserved EF

Univariate analyses assessing the relationship between cardiovascular risk factors and left ventricular diastolic dysfunction demonstrated that older age, presence of hypertension, longer duration of diabetes, and elevated HbA1c levels were significantly associated with diastolic dysfunction in patients with type 2 diabetes mellitus (Table 3).

Using LASSO regression in combination with model selection criteria including AIC, BIC, and multicollinearity assessment via variance inflation factor (VIF), a multivariable logistic regression model was constructed to identify predictors of left ventricular diastolic dysfunction. The analysis revealed that only two variables—age ≥ 60 years (OR = 2.87, *p* = 0.003) and diabetes duration of 5–10 years (OR = 3.34, *p* < 0.001)—were independently associated with diastolic dysfunction in patients with type 2 diabetes mellitus. Notably, diabetes duration > 10 years and HbA1c levels were not significantly associated with diastolic dysfunction (Table 4).

### 3.3. Risk Factors Associated with Abnormal GLS in T2DM Patients with Preserved EF

Univariate analyses assessing the association between cardiovascular risk factors and reduced GLS in patients with T2DM identified two factors significantly associated with impaired GLS: sex and left ventricular diastolic dysfunction (Table 5).

Using LASSO regression in conjunction with model selection criteria including AIC, BIC, and multicollinearity assessment via variance inflation factor (VIF), a multivariable logistic regression model was developed to identify predictors of reduced global longitudinal strain (GLS) in patients with T2DM. The analysis identified two independent predictors: male sex (OR = 2.45, *p* = 0.04) and left ventricular diastolic dysfunction (OR = 3.14, *p* = 0.010) (Table 6).

## 4. Discussion

In this study, more than half of T2DM patients with preserved EF exhibited LV diastolic dysfunction, while 13.4% demonstrated impaired GLS, underscoring the high prevalence of subclinical myocardial abnormalities in this population. Age ≥ 60 years and a diabetes duration of 5–10 years were independently associated with diastolic dysfunction, whereas male sex and the presence of diastolic dysfunction were predictors of impaired GLS. Together, these findings suggest that both diastolic and systolic abnormalities may occur early and coexist in asymptomatic diabetic patients, highlighting the clinical importance of comprehensive echocardiographic screening.

### 4.1. Prevalence of LV Diastolic Dysfunction and Reduced GLS in Patients with T2DM

In this study, LV diastolic dysfunction was observed in 53.9% of patients, indicating that more than half of individuals with T2DM and preserved EF exhibited subclinical impairment of diastolic function. Although this prevalence is relatively high, it aligns with findings from earlier investigations, such as those by Zabalgoitia et al. (2001) and Boyer et al. (2004), which identified LV diastolic dysfunction as one of the earliest and most frequent manifestations of diabetic cardiomyopathy (DCM), with reported rates ranging from 30% to 75% [18,19]. The development of DCM is driven by several overlapping mechanisms, such as AGE deposition, persistent low-grade inflammation, myocardial fibrotic remodeling, and microvascular injury, which together reduce ventricular compliance and impair relaxation [20,21]. These findings highlight the diagnostic utility of advanced echocardiographic modalities for detecting early myocardial changes in asymptomatic individuals.

Regarding structural remodeling, concentric hypertrophy was detected in 3.9% and eccentric hypertrophy in 22.8% of the study population, with a combined prevalence of 26.7% exhibiting left ventricular hypertrophy (LVH). This suggests that anatomical remodeling of the left ventricle is relatively common among patients with T2DM. The predominance of eccentric hypertrophy reflects an adaptive response to increased preload, characterized by augmented chamber volume and myocardial mass, and is often associated with diastolic dysfunction. LV remodeling is considered an intermediate stage that may precede overt systolic dysfunction [22]. Consequently, comprehensive assessment of both morphological and functional parameters is essential in both research and clinical practice.

Furthermore, reduced global longitudinal strain (GLS) was identified in 13.4% of patients, indicating that a considerable proportion of individuals with T2DM—despite the absence of overt heart failure symptoms—had already developed subclinical myocardial contractile dysfunction. This emphasizes the diagnostic value of GLS as an early marker of preclinical DCM.

GLS reflects longitudinal deformation of the myocardium, largely determined by subendocardial fibers that are highly susceptible to chronic hyperglycemia, oxidative stress, AGE build-up, and microvascular damage. These pathological changes may precede alterations in EF—which reflects volumetric systolic function—yet result in early longitudinal systolic impairment detectable through reduced GLS. The 13.4% prevalence of abnormal GLS in this cohort is consistent with international findings; for instance, Wang et al. (*JACC Cardiovasc Imaging* 2018), reported GLS abnormalities in approximately 23% of asymptomatic T2DM patients [23]. Although less prevalent than diastolic dysfunction, impaired GLS has been established as an independent predictor of heart failure and cardiovascular events in large-scale cohort studies [11,24].

Notably, reductions in GLS may appear at an early stage of diabetes, often preceding both overt symptoms and measurable declines in ejection fraction. Thus, early detection through speckle-tracking echocardiography not only enhances cardiovascular risk stratification but also provides a critical opportunity for timely therapeutic intervention aimed at mitigating the progression of diabetic cardiomyopathy.

### 4.2. Factors Associated with LV Diastolic Dysfunction in Patients with T2DM

Multivariate logistic regression analysis revealed two independent factors associated with LV diastolic dysfunction: age ≥ 60 years (OR = 2.51; 95% CI: 1.25–5.07; *p* = 0.010) and diabetes duration of 5–10 years (OR = 3.06; 95% CI: 1.46–6.40; *p* = 0.003).

#### 4.2.1. Older Age

Advancing age is a well-recognized determinant of impaired diastolic relaxation. Structural remodeling of the aging myocardium, including diffuse interstitial fibrosis, reduced elasticity, and altered cardiomyocyte–vascular interactions, leads to reduced ventricular compliance. In our cohort, patients aged 60 and above were more than twice as likely to present with LV diastolic dysfunction compared to those under 60 years. This observation is consistent with previous reports linking cardiac aging to a higher prevalence of HFpEF. For instance, a systematic review and meta-synthesis by Bouthoorn et al. (2018) reported LV diastolic dysfunction in approximately 35–48% of patients with type 2 diabetes mellitus (T2DM), indicating a high prevalence associated with aging and chronic underlying diseases [25]. A more recent study by Chee et al., conducted in Malaysia in 2021, also found that age was associated with LV diastolic dysfunction in diabetic patients [26]. Epidemiologic studies in both Western and Asian populations have shown that elderly patients with T2DM are particularly susceptible to diastolic abnormalities, likely due to the cumulative effects of aging and concomitant cardiovascular risk factors such as hypertension, dyslipidemia, and renal impairment.

#### 4.2.2. Duration of Diabetes

A diabetes duration of 5–10 years showed the strongest association with LV diastolic dysfunction, whereas the >10-year group no longer retained statistical significance in the multivariate model (OR = 1.94; *p* = 0.101). This could reflect two possibilities. First, a threshold effect: significant subclinical myocardial fibrosis and microvascular damage may accumulate after 5 years, while beyond 10 years, other factors (e.g., age, comorbidities) may dominate outcomes. Second, a selection effect: patients with diabetes > 10 years might have received more aggressive treatment or already developed overt cardiac complications (e.g., symptomatic heart failure or reduced EF), leading to their exclusion from the study and lowering the observed LV diastolic dysfunction rate. The presence of LV diastolic dysfunction after only 5 years of diabetes also supports the hypothesis that myocardial injury in diabetes begins early and progresses silently, underscoring the importance of routine echocardiographic screening, even in asymptomatic patients. Similarly, a retrospective study of 92 diabetic patients found more pronounced LV diastolic dysfunction in those with a disease duration ≥ 7 years (*p* < 0.05) [4]. The Olmsted County study (1996–2007) involving nearly 500 patients found that diabetes duration ≥ 4 years was associated with increased LVDD risk (E/e′ > 15) (OR = 1.91; *p* = 0.007) [27]. Additionally, another study on 73 T2DM patients showed that a duration of approximately 7 years of diabetes was sufficient to cause early-stage diastolic dysfunction [28]. These results suggest that an extended duration of diabetes heightens the likelihood of diastolic dysfunction, reflecting the progressive accumulation of myocardial injury over time through mechanisms such as advanced glycation end-products, oxidative stress, chronic inflammation, and microvascular impairment.

Although hypertension, elevated HbA1c, and obesity were associated with LV diastolic dysfunction in univariate analysis, they lost significance in the multivariate model. This may be because the effect of hypertension is overshadowed by the strong influence of aging−a factor closely linked to hypertension in the diabetic population. While HbA1c indicates glycemic control over recent months, myocardial injury in diabetes is a gradual process driven by chronic microvascular remodeling, fibrosis, and altered energy metabolism that develops over many years. Obesity may increase risk but was not strong enough in this sample or may exert indirect effects through blood pressure and insulin resistance.

### 4.3. Factors Associated with Reduced Global Longitudinal Strain (GLS)

In our analysis, two variables were independently associated with impaired global longitudinal strain (GLS): male sex and the presence of diastolic dysfunction.

#### 4.3.1. Sex-Related Differences

In multivariate logistic regression, male sex was independently associated with reduced GLS (OR = 2.45; 95% CI: 1.04–5.76; *p* = 0.040). This suggests that men have a 2.5-fold higher risk than women of developing longitudinal systolic dysfunction despite preserved ejection fraction. This sex disparity has been observed in other large-scale studies, where GLS was shown to be a stronger predictor of cardiovascular outcomes in men compared with women (HR = 1.14 vs. 0.99; *p* = 0.032) [24]. Other studies in HFrEF patients have also confirmed that GLS is a stronger prognostic marker in men [11,29]. Biological and hormonal factors may contribute to this difference. Men generally have greater myocardial mass, which imposes higher metabolic, and perfusion demands, potentially increasing susceptibility to microvascular dysfunction and fibrosis. In contrast, estrogen is believed to exert cardioprotective effects in women by enhancing endothelial function and attenuating myocardial remodeling. Lifestyle and behavioral factors such as higher smoking rates and suboptimal cardiovascular risk control among men may also play a role.

#### 4.3.2. Association with Diastolic Dysfunction

The finding that LV diastolic dysfunction was a strong predictor of abnormal GLS (OR = 3.14; 95% CI: 1.31–7.51; *p* = 0.010) is also clinically important. The strong link between LV diastolic dysfunction and impaired GLS in our cohort indicates that diastolic abnormalities may represent an early stage in the continuum toward systolic impairment. This finding supports the concept that DCM is a progressive disorder, where impaired relaxation and increased chamber stiffness precede subclinical reductions in longitudinal contractile performance, even when ejection fraction remains preserved. Earlier investigations in diabetic patients with preserved EF have reported comparable findings. For example, in a cohort of 177 individuals with T2DM, GLS showed a strong correlation with diastolic function parameters (E′ and E/E′), independent of age (β = 0.30; *p* < 0.001) [30]. However, other investigations suggest that GLS reduction can appear before overt diastolic dysfunction, highlighting the complex temporal relationship between these abnormalities. Ernande et al. found that in 114 diabetic patients with preserved EF, 28% had reduced GLS despite no or only early-stage LV diastolic dysfunction [31]. Liu et al. also found both GLS and E/E′ to be independent predictors of cardiovascular events in T2DM patients [17]. Therefore, reduced GLS not only reflects impaired longitudinal contractile function but also indicates early myocardial injury in the progression of diastolic dysfunction−particularly in T2DM patients with preserved EF. This highlights the importance of evaluating both LV diastolic function and longitudinal strain in early cardiovascular risk screening in T2DM patients.

In this study, the mean HbA1c level at the time of assessment did not significantly differ between the normal and abnormal GLS groups (7.42 ± 1.29 vs. 7.51 ± 1.01, *p* = 0.565). This suggests that HbA1c—a marker of average blood glucose over the past 2–3 months−is not independently associated with reduced GLS at the current time. Several hypotheses have been proposed to explain this. First, HbA1c reflects short-term glycemic control, while diabetic myocardial injury is the result of a prolonged and complex process involving microvascular remodeling, fibrosis, and impaired cellular energy metabolism−all of which require time to develop. Second is the “legacy effect,” as described in large trials such as UKPDS (United Kingdom Prospective Diabetes Study) and DCCT/EDIC (Diabetes Control and Complications Trial/Epidemiology of Diabetes Interventions and Complications), which showed that past poor glycemic control can have lasting effects on vascular and myocardial health, even if HbA1c improves later [32,33]. Moreover, GLS is highly sensitive and may be influenced by numerous factors beyond glycemia, including blood pressure, obesity, chronic inflammation, dyslipidemia, and subclinical myocardial fibrosis—all of which are not captured by HbA1c alone.

Previous studies have reported similar findings. For instance, Ernande et al. [31] found no significant association between HbA1c and GLS but observed a correlation between diabetes duration and myocardial fibrosis on cardiac MRI. This supports the notion that a single HbA1c measurement is insufficient to predict chronic myocardial damage and does not fully reflect cardiovascular burden in diabetic patients.

### 4.4. Clinical Implications

The findings of this study emphasize the importance of early cardiovascular screening in patients with T2DM, even when ejection fraction remains preserved. The high prevalence of diastolic dysfunction and impaired GLS indicates that subclinical myocardial impairment is common in this population and may precede the onset of symptomatic heart failure. Routine incorporation of advanced echocardiographic techniques, particularly speckle-tracking strain analysis, could therefore enhance the early detection of myocardial abnormalities that conventional echocardiography might overlook.

Identifying high-risk groups, such as older patients, men, and those with intermediate diabetes duration, may be especially valuable for guiding closer monitoring and earlier intervention. By detecting latent myocardial dysfunction, clinicians can initiate tailored management strategies, including aggressive risk factor control, optimization of glycemic status, and targeted pharmacologic therapies that may slow the progression of diabetic cardiomyopathy.

From a broader perspective, these results also underscore the need to integrate cardiovascular imaging into routine diabetes care. Given the silent nature of early myocardial dysfunction, reliance solely on clinical symptoms or standard EF assessment may delay diagnosis until advanced disease develops. Incorporating GLS into clinical practice may therefore improve risk stratification, aid in treatment decision-making, and ultimately reduce the burden of heart failure in patients with T2DM.

## 5. Limitations

This study has several limitations. First, its cross-sectional design precludes causal inference regarding the associations between clinical variables and subclinical myocardial dysfunction. Specifically, while we identified significant associations between factors such as age, diabetes duration, and diastolic dysfunction or reduced GLS, the study design does not allow us to determine whether these variables directly contribute to myocardial impairment or are merely correlated. Second, being a single-center study, the findings may not be generalizable to broader diabetic populations. Lastly, the absence of advanced biomarkers and cardiac MRI limited our ability to assess myocardial fibrosis or inflammation.

## 6. Conclusions

In this study of patients with T2DM and preserved ejection fraction, both left ventricular diastolic dysfunction and impaired GLS were frequently observed, highlighting the presence of subclinical myocardial involvement even in the absence of overt heart failure symptoms. Age ≥ 60 years and intermediate diabetes duration (5–10 years) were independently associated with diastolic dysfunction, whereas male sex and the coexistence of diastolic abnormalities predicted reduced GLS. These results underline the clinical relevance of advanced echocardiographic assessment in diabetic patients and suggest that incorporating GLS into routine evaluation may allow for earlier recognition of latent cardiac dysfunction. Such an approach has the potential to guide preventive interventions and mitigate the progression to symptomatic diabetic cardiomyopathy.

## Figures and Tables

**Table 1 medsci-13-00237-t001:** Characteristics of the Study Population.

			Patients (*n* = 232)
Male (*n*, %)			107 (46.1)
Age (years)			65.5 ± 10.3
Age group (*n*, %)	<60 years		60 (25.9%)
	≥60 years		172 (74.1%)
BMI (*n*, %)	Normal		76 (32.8%)
	Overweight		65 (28.0%)
	Obese		91 (39.2%)
Hypertension (*n*, %)	No		56 (24.1%)
	Yes		176 (75.9%)
Smoking (*n*, %)	No		210 (90.5%)
	Yes		22 (9.5%)
Dyslipidemia (*n*, %)	No		193 (83.2%)
	Yes		39 (16.8%)
Duration of diabetes (years)			7 (3–15)
HbA1c (%)			7.4 ± 1.2

BMI: body mass index, HbA1c: Hemoglobin A1c.

**Table 2 medsci-13-00237-t002:** Echocardiographic Characteristics.

		Patients (*n* = 232)
LVEF (%)		70.7 ± 5.6
E/A ratio		1.0 ± 0.2
E′ sept (m/s)		0.08 ± 0.02
E′ lat (m/s)		0.09 ± 0.03
E/E′ avg		8.4 ± 3.0
TRv′ (m/s)		2.08 ± 0.69
LVMI (g/m^2^)		92.7 ± 25.5
LAVI (ml/m^2^)		27.7 ± 3.2
RWT		0.33 ± 0.08
GLS		−19.5 ± 2.5
Diastolic dysfunction (*n*, %)	No	107 (46.1%)
	Yes	125 (53.9%)
LV geometry (*n*, %)	Normal	163 (70.3%)
	Concentric hypertrophy	9 (3.9%)
	Eccentric hypertrophy	53 (22.8%)
	Concentric remodeling	7 (3.0%)
GLS classification	Normal	201 (86.6%)
	Abnormal	31 (13.4%)

E′ sept: peak early diastolic tissue velocity at medial mitral annulus, E′ lat: peak early diastolic tissue velocity at lateral mitral annulus, E/A: early to late mitral filling velocity ratio; E/E′avg: average mitral-to-peak early diastolic annular ratio; GLS: global longitudinal strain; LAVI: Left Atrial Volume Index, LVMI: left ventricular mass index, RWT: relative wall thickness, TRv: peak tricuspid regurgitant velocity LV: left ventricular, LVEF: left ventricular ejection fraction.

**Table 3 medsci-13-00237-t003:** Association between risk factors and LV diastolic dysfunction.

	No Diastolic Dysfunction (*n* = 107, 46.1%) *n*, %	With Diastolic Dysfunction (*n* = 125, 53.9%) *n*, %	*p*-Value
Age < 60	42 (39.2)	18 (14.4)	<0.001 ^b^
Age ≥ 60	65 (60.8)	107 (85.6)	
Female	51 (47.7)	74 (59.2)	0.079 ^b^
Male	56 (52.3)	51 (40.8)	
Normal BMI	38 (35.5)	38 (30.4)	0.708 ^b^
Overweight	29 (27.1)	36 (28.8)	
Obese	40 (37.4)	51 (40.8)	
No Hypertension	34 (31.8)	22 (17.6)	0.012 ^b^
Hypertension	73 (68.2)	103 (82.4)	
No Smoking	98 (91.6)	112 (89.6)	0.606 ^b^
Smoking	9 (8.4)	13 (10.4)	
No Dyslipidemia	93 (86.9)	100 (80.0%)	0.160 ^b^
Dyslipidemia	14 (13.1)	25 (20.0%)	
Duration of diabetes < 5 years	48 (44.9)	28 (22.4)	0.001 ^b^
Duration of diabetes 5–10 years	26 (24.3)	52 (41.6)	
Duration of diabetes > 10 years	33 (30.8)	45 (36.0)	
HbA1c	7.2 ± 1.4	7.7 ± 1.1	<0.001 ^a^

BMI: body mass index, HbA1c: Hemoglobin A1c, ^a^: Mann–Whitney test, ^b^: Chi-squared test.

**Table 4 medsci-13-00237-t004:** Multivariate Logistic Regression for Diastolic Dysfunction.

Variable	OR	95% CI	*p*-Value
Age ≥ 60 (vs. <60)	2.51	1.25–5.07	**0.010**
Male (vs. Female)	0.61	0.33–1.14	0.123
Overweight (vs. Normal)			
Obese (vs. Normal)	1.18	0.56–2.48	0.540
Hypertension	1.31	0.65–2.65	0.452
Smoking	1.92	0.97–3.80	0.060
Dyslipidemia	2.17	0.78–6.07	0.139
Diabetes 5–10 years (vs. <5)	3.06	1.46–6.40	0.003
Diabetes > 10 years (vs. <5)	1.94	0.88–4.27	0.101
HbA1c	1.28	0.97–1.70	0.080

**Table 5 medsci-13-00237-t005:** Association between risk factors and decrease in GLS.

	Normal GLS (*n* = 201)	Abnormal GLS (*n* = 31)	*p*-Value
Age < 60	50 (24.9)	10 (32.3)	0.382 ^b^
Age ≥ 60	151 (75.1)	21 (67.7)	
Female	114 (56.7)	11 (35.5)	0.027 ^b^
Male	87 (43.3)	20 (64.5)	
Normal BMI	69 (34.3)	7 (22.6)	0.273 ^b^
Overweight	57 (28.4)	8 (25.8)	
Obese	75 (37.3)	16 (51.6)	
No Hypertension	52 (25.9)	4 (12.9)	0.116 ^b^
Hypertension	149 (74.1)	27 (87.1)	
No Smoking	185 (92.0)	25 (80.6)	0.090 ^g^
Smoking	16 (8.0)	6 (19.4)	
No Dyslipidemia	170 (84.6)	23 (74.2)	0.150 ^b^
Dyslipidemia	31 (15.4)	8 (25.8)	
No Diastolic Dysfunction	99 (49.3)	8 (25.8)	0.015 ^b^
Diastolic Dysfunction	102 (50.7)	23 (74.2)	
HbA1c (%)	7.4 ± 1.3	7.5 ± 1.0	0.565 ^a^

^a^: Mann–Whitney test, ^b^: Chi-squared test, ^g^: Fisher’s exact test.

**Table 6 medsci-13-00237-t006:** Independent factors associated with abnormal GLS in hypertensive patients by logistic regression.

	OR	95% CI	*p*-Value
Male (vs. Female)	2.45	1.04–5.76	0.040
Smoking	1.75	0.57–5.34	0.325
Diastolic Dysfunction	3.14	1.31–7.51	0.010

## Data Availability

Due to patient privacy protection and ethical restrictions, the data presented in this study are available on request from the corresponding author.

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
