# Peer review of "Subclinical Myocardial Dysfunction in Type 2 Diabetes Mellitus: Insights from Left Ventricular Diastolic Function and Global Longitudinal Strain Assessment"

_medsci, 2025, doi:10.3390/medsci13040237_

Round 1

Reviewer 1 Report

Comments and Suggestions for Authors

Dear authors,

Many thanks for submitting your work to the journal. I have attached my evaluation report.

Best regards

Comments on the Quality of English Language

Some sections of the introduction require review for grammatical and flow improvements. Long sentences must be simplified for clarity.

Author Response

 Comment 1:
“The introduction is quite informative, but at the end of the introduction and after the study’s purpose, please highlight what this study adds to the existing body of knowledge.”

Response 1:
We appreciate the reviewer’s valuable suggestion. Accordingly, we have revised the final paragraph of the Introduction to include a statement that highlights the novelty and contribution of this study to the current literature.

Comment 2:
“Please check and improve grammar/flow in some sections of the introduction. It is essential to simplify long sentences for clarity.”

Response2:
Thank you for this important observation. We have carefully reviewed the entire Introduction and revised several long or complex sentences to improve clarity and flow. We ensured that the paragraph transitions are smooth and the language is more concise. Changes have been tracked in the revised manuscript.

Comment 3:
“Please define the inclusion criteria of the study.”

Response 3:
We have now clearly stated the inclusion criteria in the Methods section

Comment 4:
“The exclusion criteria are listed, but the reasoning behind them is not explained. Please justify why you exclude patients with type 1 DM, arrhythmias, or moderate-to-severe valvular disease?”

Response 4:
We agree with the reviewer that providing a rationale enhances the clarity of the methodology. We have now added justifications for the exclusion criteria:

“Patients with type 1 diabetes mellitus were excluded due to differences in pathophysiology compared to type 2 diabetes. Those with arrhythmias were excluded because irregular rhythms may interfere with accurate echocardiographic measurements. Patients with moderate-to-severe valvular disease were excluded because valvular abnormalities can independently alter cardiac structure and function, potentially confounding the interpretation of diabetes-related changes.”

Comment 5:
“Also, a brief note on sample size calculation or power estimation would strengthen the methodology.”

Response 5:
Thank you for the suggestion. We have added a note on sample size estimation in the Methods section.

Reviewer Comment 6:
“Echocardiographic methods are well-detailed. However, specify how intra- and inter-observer variability was addressed (two physicians performed exams).”

Response 6:
We thank the reviewer for this important comment. In our study, all echocardiographic examinations were performed by two board-certified cardiologists with extensive experience in cardiac imaging. To minimize inter-observer variability, both physicians followed a standardized imaging protocol based on current echocardiographic guidelines. Key measurements were cross-checked and validated by consensus in cases of uncertainty. Although formal statistical assessment of intra- and inter-observer variability (e.g., ICC) was not performed, the consistent adherence to protocol and experience of the examiners contributed to maintaining measurement reliability.

Comment 7 & 8:
“Table 4: Please bold the p-value of 0.003, as it represents a statistically significant finding.”
“Table 6: Please bold all significant p-values (e.g., p=0.040, p=0.010). This will improve readability.”

Response:
Thank you for the suggestion. We have updated Table 4 and Table 6 accordingly. All statistically significant p-values are now bolded to enhance visual clarity and highlight key results.

Reviewer Comment 9:
“At the beginning of the Discussion, provide a synopsis of the key findings before going into the subsections.”

Response:
We agree with the reviewer and have revised the opening paragraph of the Discussion to include a concise summary of the main findings.

Reviewer 2 Report

Comments and Suggestions for Authors

This study aimed (1) to determine the prevalence of LV diastolic dysfunction and abnormal GLS in T2DM patients with preserved LVEF (≥50%), and (2) to identify the clinical and paraclinical factors associated with these forms of myocardial dysfunction.

The cross-sectional study included 232 T2DM patients (≥18 years) with EF > 50% recruited from a tertiary hospital. Standard transthoracic and speckle-tracking echocardiography were performed to assess diastolic function and GLS.

Multivariate logistic regression analysis was used to identify predictors of LV dysfunction.

The authors took into account patients' characteristics and comorbidities.

Data is presented as description, and 6 tables.

 Conclusions are aligned with the data.

Limitations of the study are briefly presented. The most important are the absence of biomarkers, and MRI evaluation.

Besides the global data, the study underlines the importance of regional data to adequate intervention strategies.

Author Response

We sincerely thank the reviewer for the thoughtful and constructive summary of our work. We are pleased that the study’s aims, methodology, and conclusions were found to be clearly aligned. We acknowledge the limitations highlighted—particularly the lack of biomarker and cardiac MRI data—which we have explicitly stated in the manuscript. These are areas we hope to explore in future studies. We also appreciate the recognition of the relevance of regional data in informing context-specific strategies, which we believe adds practical value to our findings in the management of diabetic cardiomyopathy.

Reviewer 3 Report

Comments and Suggestions for Authors

It is a nice little study. I would suggest expanding the Introduction section. Methodological section needs to be substantially expanded. I know it is a standard framework, yet the paper reads more like a proceedings paper, and I believe this is not something editors would love to see.  

Comments on the Quality of English Language

just recheck English.

Author Response

We sincerely appreciate the reviewer’s positive comments and helpful suggestions. In response, we have expanded the Introduction section to provide a broader context and strengthen the rationale for the study. Specifically, we have included additional background on subclinical cardiac dysfunction in T2DM and the clinical relevance of GLS and diastolic function assessment.

We also substantially revised and expanded the Methods section to provide more detailed information regarding the study design, inclusion and exclusion criteria, echocardiographic protocols, and statistical analysis. Our aim was to enhance the scientific rigor and clarity of the methodology to meet the expectations of a full-length research article rather than a brief proceedings paper.

Round 2

Reviewer 3 Report

Comments and Suggestions for Authors

Authors did make some effort in improving manuscript. I have no further comments.